# Glycosylated Delphinidins Decrease Chemoresistance to Temozolomide by Regulating NF-κB/MGMT Signaling in Glioblastoma

**DOI:** 10.3390/cells14030179

**Published:** 2025-01-24

**Authors:** Diego Carrillo-Beltrán, Yessica Nahuelpan, Constanza Cuevas, Karen Fabres, Pamela Silva, Jimena Zubieta, Giovanna Navarro, Juan P. Muñoz, María A. Gleisner, Flavio Salazar-Onfray, Noemi Garcia-Romero, Angel Ayuso-Sacido, Rody San Martin, Claudia Quezada-Monrás

**Affiliations:** 1Laboratorio de Virología Molecular, Instituto de Bioquímica y Microbiología, Facultad de Ciencias, Universidad Austral de Chile, Valdivia 5090000, Chile; diego.carrillo@uach.cl (D.C.-B.); nahuelpan.yessica@gmail.com (Y.N.); constanza.cuevas@alumnos.uach.cl (C.C.); karen.fabres@alumnos.uach.cl (K.F.); jimena.zubieta@alumnos.uach.cl (J.Z.); 2Laboratorio de Biología Tumoral, Instituto de Bioquímica y Microbiología, Facultad de Ciencias, Universidad Austral de Chile, Valdivia 5090000, Chile; pamela.silva01@uach.cl (P.S.); giovanna.navarro@uach.cl (G.N.); 3Millennium Institute on Immunology and Immunotherapy, Facultad de Ciencias, Universidad Austral de Chile, Valdivia 5110566, Chile; maria.gleisner@uchile.cl (M.A.G.); fsalazar@uchile.cl (F.S.-O.); 4Laboratorio de Bioquímica, Departamento de Química, Facultad de Ciencias, Universidad de Tarapacá, Arica 1000007, Chile; jpmunozb@academicos.uta.cl; 5Disciplinary Program of Immunology, Institute of Biomedical Sciences, Faculty of Medicine, Universidad de Chile, Santiago 8380453, Chile; 6Faculty of Experimental Sciences, Universidad Francisco de Vitoria, 28223 Madrid, Spain; noemi.garcia@ufv.es (N.G.-R.); ayusosa@vithas.es (A.A.-S.); 7Brain Tumour Laboratory, Fundación Vithas, Grupo Hospitales Vithas, 28043 Madrid, Spain; 8Laboratorio de Patología Molecular, Instituto de Bioquímica Y Microbiología, Universidad Austral de Chile, Valdivia 5110566, Chile

**Keywords:** delphinidins, temozolomide, glioblastoma, GSCs, chemoresistance, NF-κB, methyl guanine methyl transferase

## Abstract

Glioblastoma (GB) is a highly malignant brain tumor with a poor prognosis, with a median survival of only 14.6 months despite aggressive treatments. Resistance to chemotherapy, particularly temozolomide (TMZ), is a significant challenge. The DNA repair enzyme MGMT and glioblastoma stem cells (GSCs) often mediate this resistance. Recent studies highlight the therapeutic potential of natural compounds, particularly delphinidins, found in deep purple berries. Delphinidins are known for their ability to inhibit NF-κB signaling, a critical pathway for GB progression, chemoresistance, and MGMT expression. Our research demonstrates that glycosylated delphinidins have potential adjuvant use in the treatment of GB, offering a promising natural strategy to combat TMZ resistance. Specifically, we observed that delphinidin 3,5 di-glucoside has potent anticancer effects when used alone. Meanwhile, delphinidin 3 glucoside acted in synergy with temozolomide to decrease cell viability, highlighting its potential as an adjuvant. It also exerted a faster and more sustained inhibition of NF-κB, highlighting its potential for long-lasting therapeutic effects. These findings open new avenues for targeted therapies against glioblastoma, particularly to overcome treatment resistance.

## 1. Introduction

Glioblastoma (GB) is the central nervous system tumor that presents the highest malignancy and the worst prognosis, since the highest percentage of these tumors are fatal in the second year after diagnosis, despite being medically treated. Treatment using aggressive surgery combined with chemotherapy and radiotherapy continues to experience poor results, given the aggressiveness of the tumor [1,2,3]. Resistance to treatment is almost inevitable, and 90% of patients have an early recurrence, which often exhibits a more aggressive nature [4]. Specifically for chemotherapy, temozolomide (TMZ) is used, which is the leading and most effective drug used clinically to treat GBs [5]. TMZ is a second-generation alkylating agent that does not require hepatic metabolism for activation. Its mechanism of action depends on the conversion of TMZ at physiological pH to monomethyl triazene imidazole carboxamide (MTIC), a compound that has cytotoxic effects due to its alkylating capacity on DNA [6]. However, there are mechanisms of resistance to TMZ that inhibit the activity antitumor of the compound; in fact, it is estimated that approximately 55% of GB patients are resistant to this alkylating agent, mainly due to the response of the repair system of the methyl guanine methyl transferase (MGMT) [7]. MGMT specifically repairs the damage produced by TMZ in DNA since it eliminates the methyl group of O6 methylguanine, thus stopping the cytotoxic effect of the compound, and is considered one of the critical factors in chemoresistance to TMZ. Recently, the idea has been strengthened that a unique population of cells, called GB stem cells (GSC), are responsible for promoting resistance to TMZ because they are highly undifferentiated and tumorigenic [8]. Specifically, GSCs are those that generate tumor recurrence and, at the same time, allow the heterogeneity of tumor cells that account for chemoresistance, since they exhibit stronger resistance to conventional therapies [9]. Specifically, mesenchymal GSCs, identified with the stem cell marker CD133, have been shown to express a high level of MGMT (presenting the unmethylated MGMT promoter), and this makes them resistant to TMZ treatment, unlike pro-neural GSCs [10].

The poor treatment results against GB can be enhanced with adjuvants that improve the effectiveness of the most effective treatments, such as TMZ. In this sense, the study of new medicines is becoming increasingly necessary, and natural compounds have given a new focus in the search for agents that allow improvement or collaboration in cancer treatments. It is important to emphasize that many of the drugs used today for cancer treatment have a plant origin. Still, they can often have a toxic effect, which leads them to be subjected to different tests to verify their efficacy and safety [11]. Specifically in GB, the most widely studied natural components with antitumor effects are flavonoids, polyphenols, essential oils, terpenes, and cannabinoids [11]. One of the properties that have been investigated with greater emphasis is the sensitization of the tumor to chemotherapy. Several studies have reported that NF-κB signaling is activated in response to alkylating agents and that its overexpression is associated with chemoresistance to TMZ [12,13]. A study by Lavon et al. (2002) demonstrated a significant correlation between NF-κB activation and MGMT expression in human glial tumors, which in turn was independent of the methylation status of the enzyme gene promoter [14,15,16]. In this regard, assays with synthetic inhibitors of NF-κB in GB cells have proven to be an excellent strategy to sensitize tumor cells to TMZ, since inhibiting the pathway increases apoptosis and negatively regulates MGMT expression [17].

Among the natural compounds that have the potential to regulate signaling pathways related to chemoresistance are delphinidins, found in dark purple berries, which have been proposed as one of the main components that may have good therapeutic potential in cancer [18,19]. Most significantly, their glycosylated forms are potential chemosensitizers in glioblastoma therapy because they have the potential to regulate pathways such as PI3K/AKT, NFkB, and COX2 [20,21]. In this sense, it was determined that delphinidins could sensitize ovarian cancer cells resistant to paclitaxel through the inactivation of PI3K/AKT and ERK 1/2 signaling [19]. Furthermore, other studies observed increased apoptosis and inhibited migration, invasion, and angiogenesis in different cancer cell models [22,23,24,25]. Molecularly, they have been characterized as a potent inhibitor of NF-κB signaling, interrupting IKKα phosphorylation, IκBα degradation, and NF-κB/p65 nuclear translocation in a colon cancer model [25]. We investigated the potential TMZ-sensitizing effect of delphinidin 3 glucoside and delphinidin 3,5 di-glucoside in glioblastoma cells. Our study indicates that glycosylated delphinidins downregulate NF-κB/MGMT signaling, which sensitizes glioblastoma cells to TMZ. This is the first investigation to illustrate how glycosylated delphinidins mediate this regulation. Furthermore, it highlights the potential use of these anthocyanins as an adjuvant to TMZ therapies in glioblastoma, aiming to improve patient survival and reduce the intrinsic chemoresistance of this tumor type.

## 2. Materials and Methods

### 2.1. Cells Lines and Delphinidin Stock Preparation

U87-MG (HTB-14) cell line (Glioblastoma) and the SVGp12 (CRL-8621) cell line (Glial cells from the brain) were obtained directly from ATCC. Primary GBM38 tumor culture (GB-derived primary GSC culture, which generates TMZ-chemo-resistant mesenchymal-like GSCs), was kindly donated by Dr. Angel Ayuso-Sacido. U87-MG and SVGp12 cells were incubated in DMEM-F12 medium (Gibco, Carlsbad, CA, USA) supplemented with 10% fetal bovine serum (FBS) (Hyclone, Fremont, CA, USA. g/mL of streptomycin). For subculturing, cells were washed with 1× PBS, incubated with trypsin for 3 to 5 min, and maintained with complete DMEM-F12. GBM38 cells were cultured in M21 medium. For subculturing, washed with 1× PBS, and then treated for 10 min at 37 °C with StemPro^®^ Accutase^®^ (ThermoFisher, Waltham, MA, USA). The cell lines and the primary culture were maintained at 37 °C in a 5% CO_2_ atmosphere.

### 2.2. Viability Assays (MTS)

In independent experiments, U87-MG, SVGp12, and GBM38 were grown at 5 × 10^3^ in 96-well plates. After 24 h, cells were treated with delphinidin-3,5-di-glucoside (PHL89626 phyproof^®^, Merck, Darmstadt, Germany) or delphinidin-3-glucoside (PHL89627 phyproof^®^, Merck, Darmstadt, Germany). Both glycosylated delphinidins were dissolved in dimethyl-sulfoxide (DMSO) to prepare a 20 mM stock. Subsequently, tests were carried out with concentrations of 0, 15, 30, 60, 80, 100, 120, 180, and 240 μM of glycosylated delphinidin at 72 h. Viability was measured using CellTitter 96^®^ aqueous non-radioactive cell proliferation assay kit (Promega, Madison, WI, USA), 20 μL of the reagent was added to each well and incubated for three h. Finally, the absorbance was measured at 490 nm in a plate reader (BioTek Instruments, Inc., based in Winooski, VT, USA).

### 2.3. Luciferase Rceporter Assays

The pmir-GLO construct (Promega, Madison, WI, USA) containing the MGMT promoter downstream of the luciferase reporter gene was ordered from GenScript Biotech Corp (Piscataway, NJ, USA). The ApE-A plasmid editor software (v3.1.7, 6 November 2024) (M. Wayne Davis) was used for vector design. The pmir-GLO vector was modified by cutting the PGK promoter by digestion with BglII and ApaI. The promoter region of the MGMT gene (859 bp between base 46,036 and 46,894 on chromosome 10) was inserted upstream of the luciferase reporter gene of firefly (luc2). Renilla luciferase activity was used as a normalizer. pHAGE NF-κB-TA-LUC-UBC-GFP-W was purchased from Addgene depositor Darrell Kotton Lab [26]—transfection as stated above [27]. The Dual-Luciferase^®^ kit (Promega, Madison, WI, USA) was used to measure luciferase activity. For NF-κB activity assays, eGFP fluorescence intensity was measured in the Synergy microplate reader (BioTek Instruments, Inc., based in Winooski, VT, USA).

### 2.4. Proteome Profiler NF-κB

U87-MG cells were treated with the corresponding glycosylated delphinidin for 24 h at a concentration of 120 μM diluted in a complete medium DMEM-F12. Cells were washed and lysed with RIPA1X (Invitrogen, Carlsbad, CA, USA) containing protease and phosphatase inhibitors (Roche, Basel, Switzerland) at 4 °C. The Pierce BCA Protein Quantitation Kit (Pierce, Rockford, IL, USA) was used to determine protein concentration. Detection of proteins involved in the NF-κB signaling pathway was performed with the Proteome Profiler Human NF-κB Pathway Array kit (ARY029, R&D Systems, Minneapolis, MN, USA). The manufacturer’s instructions for the procedure were followed. The capture was carried out with the SYNGENE GBOX equipment (Synoptics Ltd., Beacon House, Nuffield Road, Cambridge, UK). Semiquantitative analysis was performed using ImageJ software version 1.52a (National Institutes of Health, Bethesda, MD, USA).

### 2.5. Reverse Transcriptase–Quantitative Polymerase Chain Reaction

RNA extraction was performed with Trizol (Invitrogen, Carlsbad, CA, USA) according to the manufacturer’s instructions, and then purification was completed with a chloroform–isopropanol procedure and 70% ethanol washes. RNA elution was performed with nuclease-free water and stored at −80 °C. Two mixtures were made for the conversion to cDNA, which, when combined, gave a final volume of 20 μL. The first mixture was made with Oligo(dT)15 Primer 25 ng/mL (Promega, Madison, WI, USA), 1 μg RNA, dNTP (Promega, Madison, WI, USA), and nuclease-free water. A 5 min incubation at 65 °C was carried out, followed by 5 min at 4 °C. The second mix was made with 1 U of RNase inhibitor (Promega, Madison, WI, USA), 5× MMLV Buffer, and 10 U of Moloney murine leukemia virus (MMLV) reverse transcriptase (Promega, Madison, WI, USA). Both mixtures were then mixed and incubated for 1 h at 37 °C. The Linegene 9640 real-time kit (Bioer, Zhejiang, Hangzhou, China) was used for cDNA amplification. The amplification conditions were as follows: 95 °C for 10 min followed by 40 cycles of denaturation at 95 °C for 15 s, annealing at 55 °C for 20 and extension at 72 °C for 20 s. The dissociation curve was performed by increasing the temperature from 70 to 90 °C in an increment of 0.5 °C. PCR mix was performed with Brilliant II SYBR^®^ Green QPCR Master Mix (Agilent Technologies, Santa Clara, CA, USA), 0.5 μM primers (Table 1), nuclease-free water, and 1 μL of cDNA in a final volume of 15 µL. β-Actin levels were used for normalization. The relative copy calculations were performed using the 2(−∆∆C(T)) method.

### 2.6. Western Blot

Protein lysates were obtained with RIPA1X buffer (Invitrogen, Carlsbad, CA, USA) containing protease and phosphatase inhibitors (Roche, Basel, Switzerland). Centrifugation was performed at 14,000× *g* for 15 min at 4 °C. Protein concentration is quantified with the Pierce BCA kit (Thermo Scientific, Rockford, IL, USA). For SDS-PAGE, 20 μg of total protein extract was loaded into 12% acrylamide-Bis gels wells. Subsequently, electroblots were performed on PVDF Hybond-P ECL membranes (Amersham, Piscataway, NJ, USA), using a pH 8.3 Tris-glycine transfer buffer (20 mM Tris, 150 mM glycine, and 20 mM methanol) and a Trans-Blot (Bio-Rad, Hercules, CA, USA). Blocking of the membranes was performed with 5% bovine serum albumin dissolved in Tris-buffered saline pH 7.6 (TBS)/0.5% Tween-20 for 1 h at RT. The membranes were then incubated overnight with the corresponding primary antibody MGMT (2739S) (Cell Signaling, Danver, MA, USA), β-Actin (SC47778) (Santa Cruz Biotechnology, Inc., Dallas, TX, USA) diluted 1:1000 in 1% BSA blocking buffer. The following day three washes were performed with TBS/Tween 20 for 10 min at RT and then incubated with HRP-conjugated secondary antibody, diluted 1:1000 in 3% BSA blocking buffer for 1 h at RT. Finally, three washes were carried out with TBS/Tween 20 for 10 min at RT and revealed with the detection reagent SuperSignal™ West Dura (Thermo Scientific, Rockford, IL, USA). The capture was carried out with the SYNGENE GBOX equipment (Synoptics Ltd., Beacon House, Nuffield Road, Cambridge, UK).

### 2.7. Chromatin Immunoprecipitation

Glioblastoma cells were treated with delphinidin or DMSO for 24 h, removed from the culture medium, and incubated with 1% formaldehyde at room temperature to crosslink protein to DNA. Subsequently, the EpiQuik Chromatin Immunoprecipitation (ChIP) kit (Epigentek, Farmingdale, NY, USA) was used following the manufacturer’s instructions. Immunoprecipitation was carried out with the following antibodies p65 NFkB (8242) (Cell Signaling, Danver, MA, USA), Non-immune IgG (1 mg/mL), Anti-RNA Polymerase II (1 mg/mL) (Epigentek, Farmingdale, NY, USA). PCR amplifications were performed on the Linegene 9640 real-time equipment (Bioer, Zhejiang, Hangzhou, China). The amplification conditions were as follows: 95 °C for 10 min followed by 35 cycles of denaturation at 95 °C for 15 s, annealing at 55 °C for 20 and extension at 72 °C for 20 s. The dissociation curve was performed by increasing the temperature from 70 to 90 °C in an increment of 0.5 °C. The primers for the chip (Table 1) are specific for the MGMT promoter and flank the NF-κB transcription factor binding site.

### 2.8. Apoptosis Assay

Before the assay, control cells were prepared for positive cell death induction, which were then divided into three independent tubes: without fluorophores, with PI only, and with Annexin V only. Experimental cells were prepared under the conditions to be studied, including a set of unstained control cells. For treatments, 500,000 cells were plated in each 60 mm dish with the following conditions: DMSO, DEL 3 120 µM, TMZ 200 µM, DEL 3.5 120 µM, DEL 3 120 µM + TMZ 200 µM, DEL 3.5 120 µM + TMZ 200 µM, BAY 11-7082 20 µM, BAY 11-7082 20 µM + TMZ 200 µM (positive control), and untreated cells. On the day of the assay, 1×Annexin V Binding Buffer was prepared by diluting 1 part of 10× buffer with 9 parts distilled water. Cells under experimental conditions were reviewed before analysis. Cells were collected after treatment, including any detached cells, and centrifuged at 600× *g* for 5 min to collect at least 5 × 10^5^ cells. The supernatant was discarded, and the cells were resuspended in 100 µL of 1× Annexin V binding buffer. Then, 5 µL of Annexin V FITC, CAT 640906, 90 µg/mL (Biolegend, San Diego, CA, USA) and 5 µL of PI solution, CAT 421301, 0.5 mg/mL (Biolegend, San Diego, CA, USA) were added. The cells were gently resuspended and incubated for 15 min at room temperature in the dark. 400 µL of Annexin V binding buffer were added to each tube. Flow cytometry was performed. The Annexin V binding buffer 10× was prepared as follows: 0.1 M HEPES, pH 7.4 with NaOH, 1.4 M NaCl, and 25 mM CaCl_2_, filtered through a 0.22 µm filter and stored at 4 °C, protected from light.

### 2.9. Statistical Analysis

Statistical analyzes were performed with the GraphPad Prism 8 program (GraphPad Software, Inc., San Diego, CA, USA). To compare the means between two groups, a Mann–Whitney test was used; for the comparison between multiple groups, one-way ANOVA and Tukey’s post hoc test were performed. All statistical analyzes were performed considering at least three biological replicates of each experiment, and statistically significant changes were assumed with a *p*-value < 0.05.

## 3. Results

### 3.1. Glycosylated Delphinidins Reduce NF-κB Activity in Glioblastoma Cells

First, the viability of U87-MG, SVGp12, and GBM38 cells was analyzed when exposed to delphinidin 3,5 di-glucoside or delphinidin 3 glucoside for 72 h at concentrations of 0, 15, 30, 60, 80, 100, 120, 180 and 240 µM. U87-MG cells showed a significant decrease in viability with 180 and 240 μM delphinidin 3,5 di-glucoside and delphinidin 3 glucoside (Figure 1a,d). SVGp12 cells showed a reduction in viability only at 240 μM of glycosylated delphinidins (Figure 1a,e). Delphinidin-3-glucoside decreased the viability of U87-MG cells by approximately 20% at 180 μM and around 30% at 240 μM. In SVG p12 cells, a 15% decrease in viability is observed with 240 μM delphinidin-3-glucoside. Delphinidin-3,5-di-glucoside decreased the viability of U87-MG cells by 70% at 180 μM and approximately 80% at 240 μM. In SVG p12 cells, an approximate 30% decrease in viability is observed with 240 μM delphinidin 3.5 di-glucoside. Interestingly, GBM38 cells did not show significant changes in cell viability with any of the doses of glycosylated delfinidins used (Figure 1c,f). In U87-MG, SVGp12, and GBM38 cells, no significant differences were observed in the viability results with doses of 0, 15, 30, 60, 80, 100, and 120 μM of glycosylated delphinidin; for this reason, in the following experiments, working doses of 15, 60, and 120 μM glycosylated delphinidin were retained.

The pHAGE NF-κB reporter plasmid was used to measure NF-κB activity, which contains four binding sites of the NF-κB transcription factor in the promoter that controls firefly luciferase expression, and eGFP fluorescence intensity was used as a normalizer. U87-MG cells were previously transfected with the pHAGE NF-κB plasmid to determine changes in NF-κB activity. Twenty-four hours post-transfection, the cells were treated for 3, 6, 24, and 48 h with DMSO, 15, 60, or 120 μM of glycosylated delphinidin (Figure 2). In addition, the drug BAY 11-7082 was used as a direct inhibition control of NF-κB (Irreversible inhibitor of IKK-alpha). As an indirect inhibition control, we used the inhibitor LY294002, which produces the blockade of PI3k/AKT signaling, a pathway recognized for inducing NF-κB activation mediated by growth factors. The results show that both delphinidin 3,5 di-glucoside and delphinidin 3 glucoside can downregulate NF-κB activity. Specifically, we noted that delphinidin 3 glucoside negatively regulates NF-κB activity from 3 h post-exposure, and it can be observed that this regulation is dose-dependent. Furthermore, we note that the downregulation produced by 120 μM delphinidin 3 glucoside resembles the downregulation produced by the inhibitor LY294002 (Figure 2a–d). Regarding delphinidin-3,5 di-glucoside, we observed that 120 μM dose is the only dose that shows significant changes at 3 h (Figure 2e). Regarding 6 and 24 h of treatment with delphinidin-3,5 di-glucoside, it can be observed that the activity of NF-κB is negatively regulated dose-dependent (Figure 2f,g). Interestingly, after 24 h, it is observed that the effect of the NF-κB activity produced by delphinidin 3,5 di-glucoside at 120 μM is like that of the BAY 11-7082 inhibitor. However, at 48 h, no effect on NF-κB activity is observed under treatment with delphinidin 3,5 di-glucoside (Figure 2h). In summary, delphinidin 3,5 di-glucoside and delphinidin-3-glucoside treatment can downregulate NF-κB activity in glioblastoma cells, varying mainly in prolongation of effect.

### 3.2. Glycosylate Delphinidins Reduce the Levels of NF-κB Pathway Proteins That Positively Correlate with MGMT Expression in Glioblastoma In Vitro

A proteomic approach to NF-κB signaling was evaluated to determine which proteins of the NF-κB pathway are regulated by glycosylated delphinidins. First, we analyzed the changes in the levels of NF-κB pathway proteins in U87-MG cells treated with delphinidin 3 glucoside at 120 μM for 24 h (Figure 3a). We detected that many signaling member proteins were downregulated with delphinidin 3 glucoside treatment compared to the DMSO control. The main downregulated proteins were STING and SHARPIN, with fold changes less than −0.5, and the main upregulated proteins were CIAP2 and BCL10, with fold changes close to 0.5. Similarly, we analyzed the changes in the levels of NF-κB pathway proteins in U87-MG cells treated with delphinidin 3,5 di-glucoside at 120 μM for 24 h (Figure 3b). As with treatment with delphinidin 3 glucoside, we detected a more significant amount of NF-κB downregulated signaling member proteins with treatment with delphinidin 3,5 di-glucoside compared to the control with DMSO. In this case, the main downregulated proteins were p53S46 and SHARPIN with fold changes less than −0.5, and the main upregulated proteins were JNK2 and IL-18RA with fold changes greater than 0.5. Considering that the levels of SHARPIN and STING proteins were the highest regulated with the delphinidin treatments at concentrations of 120 µM for 24 h, we performed a bioinformatic analysis using the GlioVis platform to evaluate the survival of patients, concerning the expression levels of both markers in glioblastoma and their possible correlation with MGMT expression. Firstly, we performed a Kaplan–Meier estimator survival analysis with 220 cases of glioblastoma (IDH WT and mutated); of these cases, 109 had high expression of SHARPIN, and 111 had low expression of the marker (Figure 3c). The median survival for cases with high SHARPIN expression was 16.2 months compared to 14.2 months for cases with low marker expression. However, in the analysis, no statistically significant difference was found. Using the Pearson method, we then analyzed a possible correlation between SHARPIN expression and MGMT expression (Figure 3d). We detected that SHARPIN expression strongly correlates with MGMT expression in glioblastoma patients. We performed the same analysis with the expression of STING (TMEM173). First, we performed a Kaplan–Meier estimator survival analysis with 220 cases of glioblastoma; 108 had high expression of TMEM173, and 112 had low expression of the marker (Figure 3e). The median survival for cases with increased expression of TMEM173 was 13.6 months compared to 19.2 months for cases with low marker expression. The differences found demonstrate that TMEM173 is related to poor patient survival. Using the Pearson method, we then analyzed a possible correlation between TMEM173 expression and MGMT expression (Figure 3f). As with SHARPIN, we detected that TMEM173 expression strongly correlates with MGMT expression in patients with glioblastoma. With the results found, we can point out that delphinidins regulate markers of the NF-κB signaling pathway that are strongly correlated with the expression of MGMT and that, specifically, STING is presented as a marker of poor prognosis of the disease that the delphinidins can regulate.

### 3.3. Glycosylated Delphinidins Reduce MGMT Expresion in Glioblastoma Cells

Considering the above results, we evaluated the MGMT transcript levels by RTqPCR in cells previously treated with delphinidin 3-glucoside or delphinidin 3,5 di-glucoside for 24 or 48 h. The treatments were administered with 15, 60, or 120 μM of each glycosylated delphinidin and an NF-κB pathway inhibition control (BAY 11-7082) known to downregulate MGMT transcript and the vehicle control DMSO. First, U87-MG cells were treated with delphinidin 3-glucoside at the above concentrations, and the MGMT transcript levels were evaluated after 24 h (Figure 4a). It is observed that delphinidin 3 glucoside negatively regulates MGMT transcript levels at 15, 60, and 120 μM; in addition, it can be observed that this regulation is statistically significant, it is independent of the dose, and the negative regulation is more important than that seen with the positive control of inhibition of the NF-κB pathway BAY 11-7082 10 μM. To check if the decrease in the MGMT transcript observed at 24 was maintained over time, we analyzed by exposing the cells to delphinidin 3-glucoside for 48 h (Figure 4b). It is observed that the exposure of delphinidin 3,5 di-glucoside for 48 h induces the negative regulation of the levels of the MGMT transcript at concentrations of 15, 60, and 120 μM; in addition, we observed that this regulation is independent of the dose and that it resembles that observed with the positive control of inhibition of the NF-κB pathway BAY 11-7082 10 μM. In parallel, we analyzed the levels of MGMT transcript when we exposed the U87-MG cells to delphinidin 3,5 di-glucoside for 24 and 48 h (Figure 4c,d). It is observed that, after 24 h of exposure to delphinidin 3,5 di-glucoside, it negatively regulates the levels of the MGMT transcript at 15, 60, and 120 μM; in addition, it can be observed that this regulation is statistically significant and that the higher doses of the compound have lesser effects on the levels of the transcript. After 48 h, exposure to delphinidin 3,5 di-glucoside at concentrations of 15 and 60 μM induces negative regulation of MGMT transcript levels. In addition, we observed that the higher doses of this delphinidin seem less effective. in the regulation of the transcript. We analyzed MGMT protein levels in U87-MG cells exposed to delphinidin 3-glucoside or delphinidin 3,5 di-glucoside for 24 or 48 h using western blot (Figure 4e,f). The data indicate that delphinidin 3 glucoside negatively regulates the levels of MGMT only after 48 h post-exposure; instead, delphinidin 3,5 di-glucoside achieves a decrease in the levels of MGMT protein at 24 h of exposure. However, at 48 h, it was observed that doses of 120 µm of delphinidin 3,5 di-glucoside did not induce a negative regulation of the MGMT protein. Together, these data indicate that glycosylated delphinidins can regulate MGMT’s transcript and protein levels after 24 and 48 h of exposure.

### 3.4. Glycosylated Delphinidins Negatively Regulate MGMT Promoter Activity in Glioblastoma Cells Through Regulation of the NF-κB Pathway

To verify that the regulation of MGMT transcripts is mediated by a regulation of the transcriptional activity of the promoter, we modified the dual-luciferase reporter vector pmir-GLO to leave the regulatory region and the MGMT promoter in charge of the transcription of the LUC2 gene. The normalizing control we used is the renilla luciferase reporter in the vector. First, the U87.MG cells previously transfected with the pmir-GLO MGMT promoter vector were treated with delphinidin 3-glucoside or delphinidin 3,5-di-glucoside for 24 or 48 h. The treatments were performed with 15, 60, or 120 μM of each glycosylated delphinidin. In addition, an inhibition control of the NF-κB pathway BAY 11-7082 was added in each experiment. The results show that, at 24 h, delphinidin 3-glucoside negatively regulates the levels of MGMT promoter activity at 15, 60, and 120 μM, and it can be observed that this regulation is significant, independent of the dose and the negative regulation, like that seen with the positive control of inhibition of the NF-κB pathway BAY 11-7082 (Figure 5a). It is observed that, at 48 h, the behavior in the activity of the MGMT promoter is similar, showing that the negative regulation of the activity induced by delphinidin 3-glucoside is maintained over time and that it behaves similarly to the positive control of inhibition of the NF-κB pathway BAY 11-7082 (Figure 5b). On the other hand, it is observed that delphinidin 3,5-di-glucoside reduces the activity of the MGMT promoter at 24 h in a dose-dependent manner, achieving effects similar to BAY 11-7082 with 120 μM of delphinidin 3,5-di-glucoside (Figure 5c). However, the effect of delphinidin 3,5-di-glucoside is diminished at 48 h (Figure 5d). We then performed a CHIP assay with the p65/Rel-A antibody to observe the impact of glycosylated delphinins in the regulation of the binding of this transcription factor to the MGMT promoter region (Figure 5d). Interestingly, we identified that only delphinidin 3 glucoside can regulate the binding of p65/Rel-A to the MGMT promoter region. Through these assays, we observed that glycosylated delphinidins can negatively regulate the activity of the MGMT promoter. In addition, a more significant and longer-lasting effect was observed with delphinidin 3 glucoside, which would partially restrict the activity of the MGMT promoter by negative regulation of the binding of p65/Rel-A to this promoter.

### 3.5. Glycosylated Delphinidins Sensitize Glioblastoma Cells to Temozolomide

Since we observed that glycosylated delphinidins downregulate the primary direct repair mechanism of TMZ damage, we analyzed the ability of delphinidins to sensitize U87-MG cells to the chemotherapeutic compound TMZ. For this purpose, we treated the cells with nontoxic doses of glycosylated delphinidin to evaluate the combinatorial effect of TMZ with delphinidin 3-glucoside or delphinidin 3,5-di-glucoside. The doses of TMZ used were 100, 200, 400, and 800 μM, and the doses of glycosylated delphinidin were 15, 60, and 120 μM. BAY11-7082 was used as a positive sensitization control, and DMSO was used as a vehicle control. First, using the MTS assay, we evaluated the combined effect of delphinidin 3-glucoside and TMZ at 48 h in U87-MG cells (Figure 6a). In addition, we made a heat map to visualize the changes in viability and added the statistics in each representative box of the condition (Figure 6b). It is observed that delphinidin 3-glucoside at 120 μM has remarkable sensitizing effects on U87-MG cells when combined with TMZ concentrations that resemble those physiologically achieved in patients, such as 100 and 200 μM of the compound. However, it is observed that BAY 11-7082 at 20 μM has a more significant sensitizing effect than delphinidin 3-glucoside at 120 μM. Using the MTS assay, we then evaluated the combined effect of delphinidin 3,5 di-glucoside and TMZ at 48 h on U87-MG cells (Figure 6c). As in the previous results, we performed a heat map to visualize the changes in viability and added the statistics in each representative box of the condition (Figure 6d). It is observed that delphinidin 3,5 di-glucoside at the different concentrations used has minor sensitizing effects on U87-MG cells when combined with TMZ concentrations that resemble those physiologically achieved in patients, such as 100 and 200 μM of the compound. To determine whether these changes were attributable to increased apoptosis, we performed annexin V/propidium iodide assay in cells treated for 48 h with DMSO control, 200 μM TMZ, 20 μM BAY11-7082, 120 μM delphinidin, 120 μM delphinidin 3 glucoside 3,5 di-glucoside, or combined treatments (Figure 6e). We observed that combined treatment of 200 μM TMZ with 120 μM delphinidin 3-glucoside slightly increased total apoptosis compared to 200 μM TMZ alone. Furthermore, it was observed that the combinatorial treatment of 200 μM TMZ with 20 μM BAY11-7082 had a more significant effect than that produced by the combined treatment of 200 μM TMZ with 120 μM delphinidin 3-glucoside in U87-MG cells. Regarding delphinidin 3,5-di-glucoside, we observed that the combined treatment with 200 μM TMZ + 120 μM delphinidin 3,5 di-glucoside slightly increased total apoptosis compared to treatment with 200 μM TMZ. The data were presented in heat map format under the conditions described above (Figure 6f). In summary, these data indicate that glycosylated delphinidins sensitize glioblastoma cells to TMZ, regulating at least in part a slight increase in apoptosis. Therefore, glycosylated delphinidins could have potential use as a TMZ-sensitizing adjuvant compound in differentiated glioblastoma cells.

## 4. Discussion

The results demonstrate that glycosylated delphinidin, specifically delphinidin 3 glucoside and delphinidin 3,5 di-glucoside, can reduce cell viability and NF-κB activity in glioblastoma cells. This finding is relevant, since NF-κB plays a crucial role in the progression and resistance to therapies in glioblastoma, promoting cell proliferation and resistance to apoptosis in tumor cells [28]. First, when evaluating cell viability, it was observed that both delphinidin 3 glucoside and delphinidin 3,5 di-glucoside significantly reduce the viability of U87-MG cells at high doses (180 and 240 μM). Furthermore, it is worth mentioning that the observed effects on cell viability may be associated with the compound itself and also with its metabolic forms, such as gallic acid, that can induce apoptosis in other types of cancer [29,30]. In contrast, SVGp12 and GBM38 cells were less sensitive to these compounds, with a decrease in viability only at high doses and to a lesser extent. This difference in sensitivity between cell lines might be related to the specific characteristics of each line, considering that SVGp12 is a non-tumor cell line and GBM38 is a primary tumor culture of mesenchymal GSCs cells that are highly resistant to conventional therapies [31]. Our focus in developing this research was to evaluate the sensitizing effects of molecular changes that glycosylated delphinidins could offer. For this reason, we continued evaluating these metabolites at non-toxic doses for the cell. Regarding NF-κB activity, assessment using a luciferase reporter vector controlled by a promoter activated by NF-κB showed a significant reduction after treatment with delphinidin 3-glucoside and delphinidin 3,5-glucoside. In particular, delphinidin 3-glucoside downregulated NF-κB activity in a dose-dependent manner from 3 h post-treatment, while delphinine 3,5-glucoside showed a more pronounced effect at 24 h at concentrations of 120 μM. This result suggests that delphinidin 3 glucoside may have a faster and longer-lasting effect on regulating NF-κB activity, while delphin 3,5 glucoside has a slower and shorter-lasting effect. This difference can be mainly explained by factors related to their molecular structure, presenting differences in the bioavailability and metabolism of both compounds. Delphinidin 3,5 di-glucoside has two glucose molecules attached to it, creating a more significant steric hindrance [32], which could hinder its potential direct interaction with critical components of the NF-κB pathway, such as IKK proteins or another pathway component [25]. In contrast, delphinidin-3-glucoside, having a more straightforward structure, could interact more effectively with these cellular components. Previous studies have shown that di-glucosides are less quickly taken up by cells and tend to be excreted unmetabolized to bioactive products, such as gallic acid or protocatechuic acid [33], which have anti-inflammatory properties and can act as NF-κB inhibitors [34,35]. In contrast, mono-glucosides, such as delphinidin-3-glucoside, are more likely to be metabolized to these bioactive derivatives, contributing to their ability to inhibit NF-κB. Downregulation of NF-κB by these compounds might offer a therapeutic alternative in managing glioblastomas with chronic NF-κB activation, a factor associated with poor prognosis in these patients [36]. Proteomic evaluation of the NF-κB signaling pathway in cells treated with delphinidin 3-glucoside and delphinidin 3,5 di-glucoside revealed the regulation of several key members of this pathway. In the case of treatment with delphinidine 3-glucoside, a significant reduction in STING and SHARPIN levels was observed, while CIAP2 and BCL10 showed an increase. On the other hand, in treatment with delphinidin 3,5 di-glucoside, p53S46, and SHARPIN were reduced, with an increase in JNK2 and IL-18RA. These results suggest that both delphinidine glycosylates have the ability to differentially modulate NF-κB signaling, possibly through mechanisms dependent on the specific interactions of each compound with components of the NF-κB pathway or pathways that connect to it [37]. The results show that delphinidin 3-glucoside upregulates JNK2 by about 0.2-fold; interestingly, we observed that delphinidin 3,5-di-glucoside upregulates JNK2 by about 1-fold. These findings reveal the differential regulatory potential of both glycosylated definitions. Furthermore, they highlight that JNK2 upregulation is associated with pro-apoptotic mechanisms when the NF-κB pathway is inactivated [38,39]. Specifically, the differences found in the expression of JNK1/2 and JNK2 in the treatments of each of the delphinidins can be attributed to a possible direct interaction with the proteins, which has not been proven; here the enormous difference in the structure of both proteins acquires a special interest. In addition, it should be considered that JNK2 is considered a tumor suppressor and has been shown to have an opposite role to that of JNK1 in tumor tissue [40].The finding of a significant correlation between SHARPIN and STING expression and MGMT expression in glioblastoma is of particular interest since MGMT protein is a marker of resistance to TMZ therapy [7,41]. The data suggest that delphinidins may indirectly influence therapy resistance by regulating NF-κB and MGMT expression. In this regard, STING is a marker of poor prognosis, since high levels of this protein expression were associated with lower survival in patients with glioblastoma. The reduction of STING and SHARPIN by delphinidins 3-glucosides and delphinidins 3,5 di-glucoside, respectively, may indicate the functionality of this type of anthocyanins. However, to establish these proteins as markers in the treatment with glycosylated delphinins, more experiments are required. Regarding MGMT expression analyses, RT-qPCR and western blot results indicated that both delphinidine 3-glucoside and delphinidine 3,5 di-glucoside are effective in reducing MGMT transcript and protein levels, especially after 48 h of treatment. The reduction of MGMT is significant at all doses tested. It shows a similar or greater downregulation than that induced by the NF-κB pathway inhibitor BAY 11-7082, suggesting that these compounds could have a robust and sustained effect on MGMT expression [42]. This is particularly important, since MGMT inhibition has been associated with increased sensitivity to temozolomide and better response to treatment in glioblastoma [7,41,42]. Interestingly, the inhibitory activity on MGMT is less pronounced at high doses of delphinidine 3,5 di-glucoside after 48 h of exposure, which could indicate an adaptation of the cells to high concentrations or a concentration-dependent regulation effect [43]. This aspect could be relevant in the design of future combination therapies, where moderate doses of delphinidine could maximize MGMT inhibition without inducing adaptive responses in tumor cells. Promoter activity assays and chromatin immunoprecipitation (ChIP) experiments showed that both forms of glycosylated delphinidine can reduce MGMT promoter activity by inhibiting p65 NF-κB binding to the promoter of this gene. The effect of glycosylated delphinidins on the promoter was maintained after 24 and 48 h of treatment. This was comparable to the effect on MGMGT expression with the NF-κB inhibitor BAY 11-7082 [42]. These results provide evidence that the downregulation of MGMT by delphinidins might be mediated by a decrease in NF-κB activity, which is consistent with previous studies suggesting a regulatory role of NF-κB on MGMT expression in glioblastoma [16]. The fact that delphinidine 3,5-di-glucoside loses efficacy after 48 h might indicate a temporary regulation of NF-κB, where the activation of compensatory pathways in the tumor cell could restore MGMT promoter activity [44]. Future studies could explore this behavior to optimize delphinidin in combination with compensatory pathway inhibitors or intermittent therapies to maximize their effect [45]. Combination experiments with temozolomide revealed a clear additive effect between delphinidin-3-glucoside and temozolomide, significantly decreasing cell viability, which did not occur with the combination of temozolomide and delphinidin-3,5-di-glucoside. These results suggest that delphinidin-3-glucoside could be a more effective adjuvant in temozolomide-based therapies if its effects on the temozolomide resistance pathway associated with MGMT and NF-κB are also considered [46]. In this research, we did not investigate the post-transcriptional regulations regulated by glycosylated delphinidins; for future research, it would be interesting to evaluate the behavior of miRNAs under treatment with delphinidin 3-glucoside and delphinidin 3,5-di-glucoside. Specifically, Murata et al. found that delphinidin aglycone, when administered orally at a dose of 20 mg/kg daily for one week in mice, induced a differential expression profile of miRNAs in plasma, related mainly to Hippo signaling and TGF-beta signaling [47]. Furthermore, evaluating the dual use of miRNA with glycosylated delphinins to enhance antitumor mechanisms would be interesting. In this regard, Chakrabarti et al. have demonstrated that the effect of delphinidin aglycone in collaboration with a miR-137 mimic enhances antitumor mechanisms in glioblastoma cells [48].

In summary, glycosylated delphinidines, especially delphinidine 3,5-di-glucoside, significantly reduce cell viability and inhibit the NF-κB signaling pathway in glioblastoma cells. Furthermore, these molecules reduce MGMT expression, which might increase the sensitivity of glioblastoma cells to alkylating agents, such as temozolomide. These findings suggest that glycosylated delphinidines have potential as adjuvants in the treatment of glioblastoma, and their role in regulating NF-κB and MGMT activity opens new avenues for the development of targeted therapies (Figure 7). Additional research is required to understand the underlying mechanisms and to determine these delphinidines’ effectiveness in other tumor cell types and their long-term therapeutic potential.

## Figures and Tables

**Figure 1 cells-14-00179-f001:**
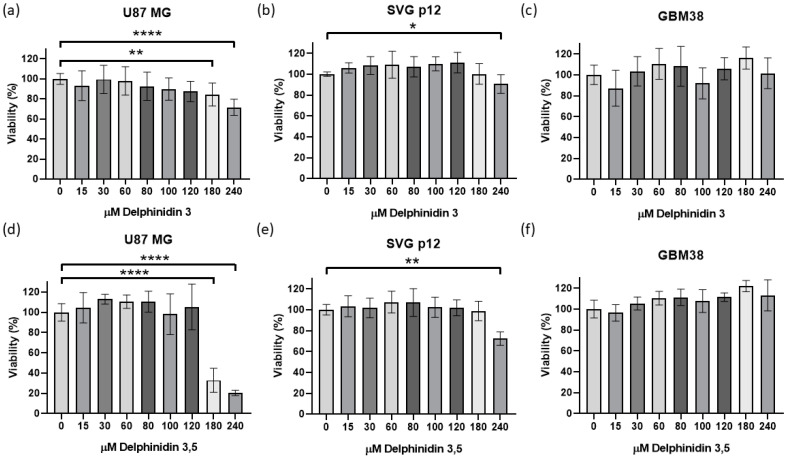
Cytotoxicity assay of glycosylated delphinidins exposed to different concentrations in tumor and non-tumor glial cells. MTS assay to evaluate the viability of cells when exposed for 72 h to concentrations of 0, 15, 30, 60, 80, 100, 120, 180, and 240 μM of delphinidin 3 glucoside or delphinidin 3,5 di-glucoside. (**a**) U87-MG exposed to delphinidin 3 glucoside. (**b**) SVG-p12 exposed to delphinidin 3 glucoside. (**c**) GBM38 exposed to delphinidin 3 glucoside. (**d**) U87-MG exposed to delphinidin 3,5 di-glucoside. (**e**) SVG-p12 exposed to delphinidin 3,5 di-glucoside. (**f**) GBM38 exposed to delphinidin 3,5 di-glucoside. Data are presented as the mean ± standard deviation (SD); * *p* < 0.05; ** *p* < 0.01 and **** *p* < 0.0001.

**Figure 2 cells-14-00179-f002:**
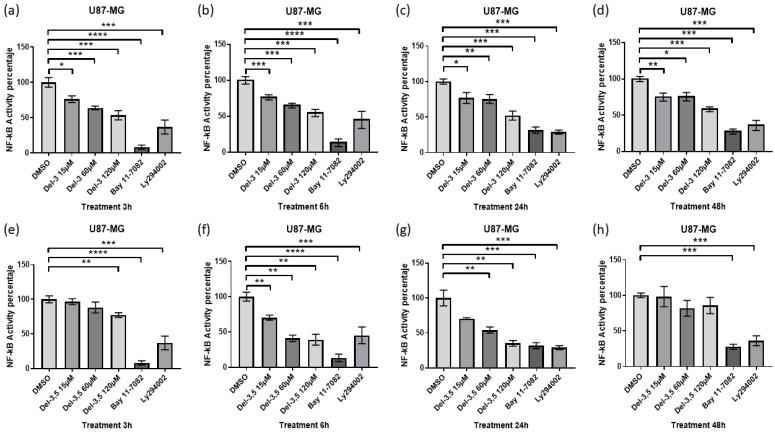
Glycosylated delphinidins reduce NF-κB activity in glioblastoma cells. (**a**) Luciferase activity normalized to GFP fluorescence intensity in U87-MG cells transfected with the pHAGE/NF-κB reporter vector treated with delphinidin-3-glucoside at concentrations of 15, 60, and 120 μM or with controls Bay117082 10 nM or LY294002 10 nM for 3 h. (**b**) 6 h. (**c**) 24 h. (**d**) 48 h. (**e**) Luciferase activity normalized to GFP fluorescence intensity in U87-MG cells transfected with the pHAGE/NF-κB reporter vector treated with delphinidin-3,5-di-glucoside at concentrations of 15, 60, and 120 μM or with the controls Bay117082 10 nM or LY294002 10 nM for 3 h. (**f**) 6 h. (**g**) 24 h. (**h**) 48 h. Data are presented as the mean ± standard deviation (SD); * *p* < 0.05; ** *p* < 0.01; *** *p* < 0.001 and **** *p* < 0.0001.

**Figure 3 cells-14-00179-f003:**
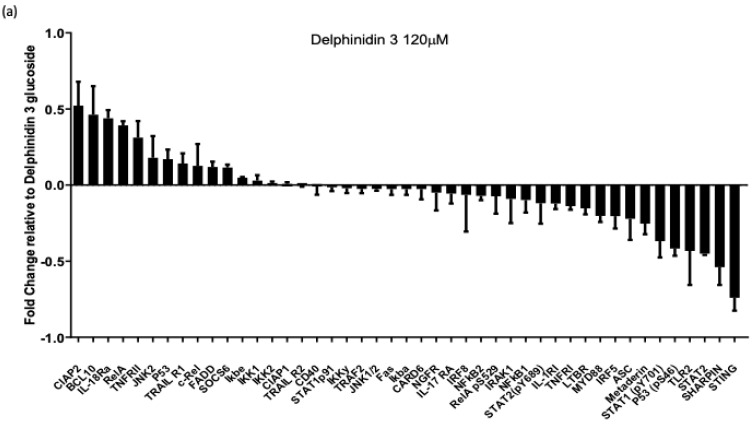
Glycosylate delphinidins reduce the levels of NF-κB pathway proteins that positively correlate with MGMT expression in glioblastoma in vitro. Protein array of the NF-κB signaling pathway performed in U87-MG cells treated with delphinidin 3 glucoside or delphinidin 3,5 di-glucoside at 120 µM for 24 h; the graph demonstrates the fold change compared to the control with DMSO. (**a**) U87-MG cells treated with delphinidin 3 glucoside. (**b**) U87-MG cells treated with delphinidin 3,5 di-glucoside. (**c**) Survival analysis with Kaplan–Meier plot of 220 glioblastoma cases; the red line represents tumors with high levels of SHARPIN, and the blue line represents tumors with low levels of the marker; the graph reports median survival and Hazard Ratio (HR) along with statistical significance. (**d**) Pearson correlation between SHARPIN and MGMT expression levels. (**e**) Survival analysis with Kaplan–Meier plot of 220 glioblastoma cases; the red line represents tumors with high levels of TMEM173, and the blue line represents tumors with low levels of the marker; the graph reports median survival and Hazard Ratio (HR) along with statistical significance. (**f**) Pearson correlation between TMEM173 and MGMT expression levels. Data are presented as the mean ± standard deviation (SD).

**Figure 4 cells-14-00179-f004:**
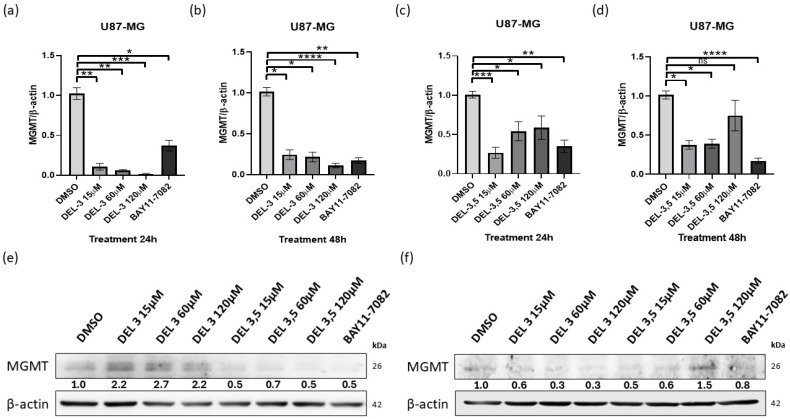
Glycosylate delphinidins reduce transcript and protein levels of MGMT in glioblastoma cells. RTqPCR was performed to evaluate MGMT transcript levels in U87-MG cells previously exposed to delphinidin 3 glucoside or delphinidin 3,5 di-glucoside at concentrations of 15, 60, and 120 μM for 24 or 48 h, BAY11-7082 was used at 10 μM. (**a**) RTqPCR of treatment delphinidin 3-glucoside for 24 h. (**b**) RTqPCR of treatment delphinidin 3-glucoside for 48 h. (**c**) RTqPCR of treatment delphinidin 3,5 di-glucoside for 24 h. (**d**) RTqPCR of treatment delphinidin 3,5 di-glucoside for 48 h. (**e**) WB was performed to assess MGMT protein levels in U87-MG cells when exposed to delphinidin 3-glucoside or delphinidin 3,5 di-glucoside for 24 h; the control BAY 11-7082 was used at 10 μM. (**f**) WB was performed to assess MGMT protein levels in U87-MG cells when exposed to delphinidin 3 glucoside or delphinidin 3,5 di-glucoside for 48 h. β-actin transcript was used as an endogenous control in the RTqPCR. Data are presented as mean ± standard deviation (SD); * *p* < 0.05; ** *p* < 0.01; *** *p* < 0.001 and **** *p* < 0.0001.

**Figure 5 cells-14-00179-f005:**
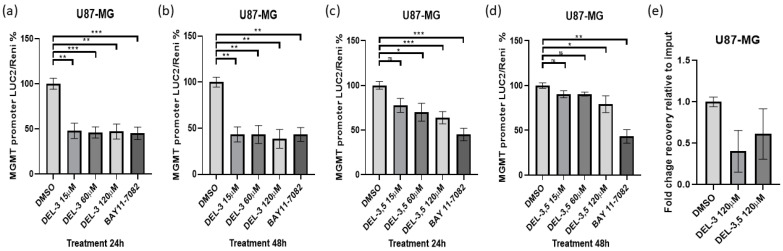
Glycosylated delphinidins negatively regulate MGMT promoter activity in glioblastoma cells. (**a**) The luciferase reporter assay was performed with the pmir-GlO MGMT promoter vector. To perform the assay, the U87-MG cells were previously transfected with the vector for 24 h; then treatments were carried out with delphinidin 3-glucoside at concentrations of 15, 60, and 120 μM for 24 h; BAY11-7082 was used as a positive regulation control. The data were normalized with the activity of renilla luciferase (**b**). The previous assay was performed with exposure to delphinidins for 48 h. (**c**) The activity of the MGMT promoter was measured when U87-MG cells were exposed to delphinidin 3,5 di-glucoside at concentrations of 15, 60, and 120 μM for 24 h. (**d**) Assay to measure MGMT promoter activity when cells are exposed for 48 h to delphinidin 3,5 di-glucoside. (**e**) Chromatin immunoprecipitation assay with anti p65/Rel-A antibody in U87.MG cells were exposed for 24 h to 120 μM of glycosylated delphinidins; the immuno-precipitate obtained with anti-RNA Polymerase II amplified with primers from the GAPDH promoter region was used as a normalizer. Data are presented as the mean ± standard deviation (SD); * *p* < 0.05; ** *p* < 0.01; and *** *p* < 0.001.

**Figure 6 cells-14-00179-f006:**
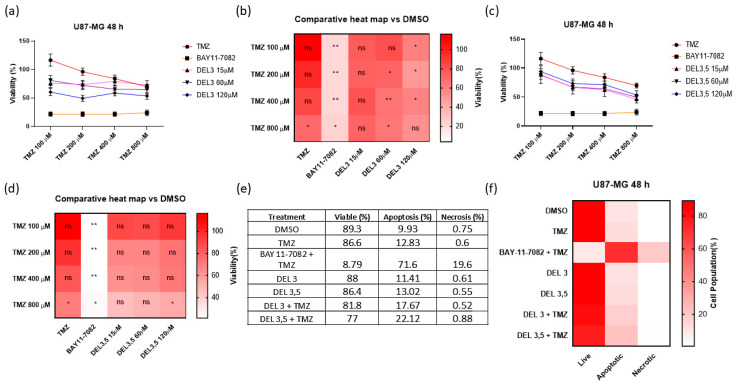
Sensitizing capacity of glycosylated delphinidins to the drug TMZ in glioblastoma cells. (**a**) MTS assay to assess the viability of U87-MG cells when exposed for 48 h to the combined treatments of TMZ at 100, 200, 400 and 800 μM, with delphinidin 3 glucoside doses at 15, 60 and 120 μM. BAY11-7082 20 μM was used as a positive sensitization control and DMSO as a vehicle control. Data were adjusted to percentages taking DMSO treatment as 100%. (**b**) Heat map of the previous MTS assessing delphinidin 3-glucoside-induced TMZ sensitization, including statistical analysis in each quadrant. (**c**) MTS assay to assess the viability of U87-MG cells when exposed for 48 h to the combined treatments of TMZ at 100, 200, 400, and 800 μM, with delphinidin 3,5 di-glucoside doses at 15, 60, and 120 µM. BAY11-7082 20 μM was used as a positive sensitization control, and DMSO was used as a vehicle control. Data were adjusted to percentages taking DMSO treatment as 100%. (**d**) Heatmap of the above MTS assessing the sensitization to TMZ induced by delphinidin 3,5 di-glucoside, including statistical analysis in each quadrant. (**e**) Summary table of percentages of apoptosis data obtained with the 48-h combinatorial treatment with 200 μM TMZ, 120 μM delphinidin 3 glucoside and 120 μM delphinidin 3,5 di-glucoside. 20 μM BAY11-7082 was used as a positive sensitization control and DMSO as a vehicle control. (**f**) Heat map of the previous apoptosis assay. Data are presented as the mean ± standard deviation (SD); * *p* < 0.05 and ** *p* < 0.01.

**Figure 7 cells-14-00179-f007:**
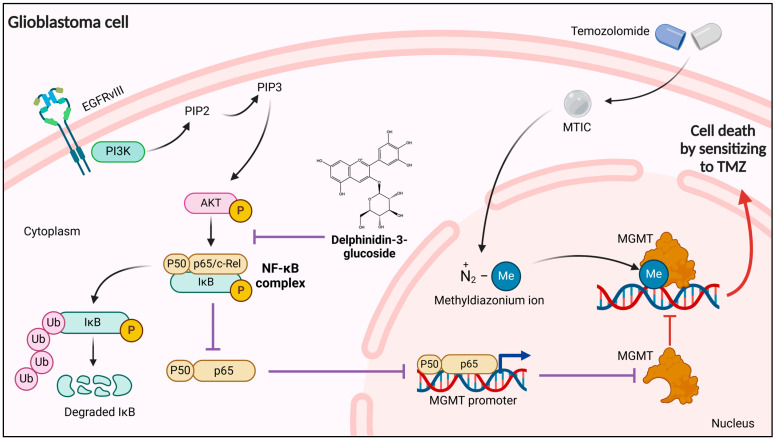
A proposed model of capacity of glycosylated delphinidins to decrease chemoresistance to Temozolomide by regulating NF-κB/MGMT (Created in https://BioRender.com (accessed on 12 December 2024)).

**Table 1 cells-14-00179-t001:** Primer list.

Primer	Forward 5′-3′	Reverse 5′-3′
CHIP MGMT	AGGACCGGGATTCTCACTAA	AGCCGACCTGAGAAA
MGMT	GCAATTAGCAGCCCTGGCA	CACTCTGTGGCACGGGA
β-Actin	GAGCACAGAGCCTCGCCTTT	CACGATGGAGGGGAAGAC

## Data Availability

Data are contained within the article and Appendix A.

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
