# Peer review of "Glycosylated Delphinidins Decrease Chemoresistance to Temozolomide by Regulating NF-κB/MGMT Signaling in Glioblastoma"

_cells, 2025, doi:10.3390/cells14030179_

Round 1

Reviewer 1 Report

Comments and Suggestions for Authors

This manuscript titled “ Glycosylated delphinidins decrease chemoresistance to Temozolomide by regulating NF-kB/MGMT signaling in glioblastoma” The comments for this manuscript are as follows:

This manuscript is well written in the section of introduction, but in the results, the readers are not satisfied with the results. For example, in Figure 3 (a) and (b), why is there a big difference between delphinidin 3 glucoside and delphinidin 3,5 diglucoside, such as JNK 1/2 and JNK 2, etc. The authors did not explain? The difference in structure is caused by the huge difference between the two proteins should be an area where the author can make good use of it. However, the reviewer did not see a relevant explanation.

In addition, in (e) and (f) of figure 4, not only the MGMT protein is unclear, but also its beta-actin shows different levels. Please explain it or redo the experiment again.

Previous studies have found that in glioblastoma, the expression of miR-137 is a tumor suppressor miRNA. I wonder why the author did not explore this factor ?

Chakrabarti M., Ray S.K. Direct transfection of miR-137 mimics is more effective than DNA demethylation of miR-137 promoter to augment anti-tumor mechanisms of delphinidin in human glioblastoma U87MG and LN18 cells. Gene. 2015;573:141–152. doi: 10.1016/j.gene.2015.07.034. 

My suggestion is major revision.

Author Response

Reviewer 1

This manuscript is well written in the section of introduction, but in the results, the readers are not satisfied with the results. For example, in Figure 3 (a) and (b), why is there a big difference between delphinidin 3 glucoside and delphinidin 3,5 diglucoside, such as JNK 1/2 and JNK 2, etc. The authors did not explain? The difference in structure is caused by the huge difference between the two proteins should be an area where the author can make good use of it. However, the reviewer did not see a relevant explanation.

R: We are tremendously grateful for this suggestion; we investigated this topic and included a specific section addressing these changes in JNK1 and JNK2 with the different delphinidins.

In addition, in (e) and (f) of figure 4, not only the MGMT protein is unclear, but also its beta-actin shows different levels. Please explain it or redo the experiment again.

R: Thanks for this review. We added a better-resolution figure and included the densitometer analysis at the bottom of the MGMT WB

Previous studies have found that in glioblastoma, the expression of miR-137 is a tumor suppressor miRNA. I wonder why the author did not explore this factor ?

R: We appreciate this comment. At the moment, we have not focused completely on post-transcriptional regulations, specifically those mediated by miRNA, but we are especially interested in the upcoming research that continues with glycosylated delphinidins. We add a brief discussion on this type of regulatory mechanism.

Chakrabarti M., Ray S.K. Direct transfection of miR-137 mimics is more effective than DNA demethylation of miR-137 promoter to augment anti-tumor mechanisms of delphinidin in human glioblastoma U87MG and LN18 cells. Gene. 2015;573:141–152. doi: 10.1016/j.gene.2015.07.034.

R: We discussed this paper in the discussion

Reviewer 2 Report

Comments and Suggestions for Authors

This manuscript provides relevant insights into how glycosylated delphinidins may enhance glioblastoma treatment when combined with temozolomide (TMZ). The overall methodology and findings are encouraging, particularly the demonstration of NF-κB/MGMT pathway downregulation to overcome drug resistance. However, a few revisions would help make the work more concise and persuasive:

In Section introduction

1.        Consider streamlining survival and recurrence details of glioblastoma to avoid redundancy. Condensing these statistics into one paragraph would focus the narrative.

2.        Replace colloquial phrases like “Well, it should be noted...” with a more formal expression (e.g., “It is important to emphasize...”).

3.        Provide a concise overview of glycosylated delphinidins early on, clarifying their proposed benefit in GB therapy.

4.        The importance of NF-κB is repeated multiple times. Summarizing it effectively at the first mention and referencing it briefly later may improve readability.

5.        Conclude the introduction with a brief statement outlining the significance and main objectives of the study.

In section Results and Discussion

6.        Streamline the descriptions of Figures (especially Figs 1, 3, and 4) to avoid excessive repetition.

7.        Enhance the resolution of Figures 3 and 6 for clearer visualization and improved presentation.

8.        In Section 3.2, clarify how proteins were statistically selected as biomarkers. If the differences are not statistically significant, it may be premature to propose SHARPIN as a biomarker.

9.        Provide quantitative (densitometric) analysis for the Western blots in Figure 4 to strengthen the protein expression results.

10.    When discussing delphinidin 3,5-diglucoside losing efficacy after 48 hours, please include references to relevant literature that support the notion of compensatory pathways.

Overall, the manuscript addresses a pressing need in glioblastoma therapy. With these refinements, it has the potential to be an important contribution to the field.

Author Response

Reviewer 2

This manuscript provides relevant insights into how glycosylated delphinidins may enhance glioblastoma treatment when combined with temozolomide (TMZ). The overall methodology and findings are encouraging, particularly the demonstration of NF-κB/MGMT pathway downregulation to overcome drug resistance. However, a few revisions would help make the work more concise and persuasive:

In Section introduction

  1. Consider streamlining survival and recurrence details of glioblastoma to avoid redundancy. Condensing these statistics into one paragraph would focus the narrative.

R: Thanks for this suggestion, this section was reduced

  1. Replace colloquial phrases like “Well, it should be noted...” with a more formal expression (e.g., “It is important to emphasize...”).

R: The sentences were modified as indicated.

  1. Provide a concise overview of glycosylated delphinidins early on, clarifying their proposed benefit in GB therapy.

R: This section was added in the introduction

  1. The importance of NF-κB is repeated multiple times. Summarizing it effectively at the first mention and referencing it briefly later may improve readability.

R: This section has been summarized.

  1. Conclude the introduction with a brief statement outlining the significance and main objectives of the study.

R: We appreciate this suggestion; this section has been improved.

In section Results and Discussion

  1. Streamline the descriptions of Figures (especially Figs 1, 3, and 4) to avoid excessive repetition.

R: Figure descriptions have been modified

  1. Enhance the resolution of Figures 3 and 6 for clearer visualization and improved presentation.
  2. The figures were modified
  3. In Section 3.2, clarify how proteins were statistically selected as biomarkers. If the differences are not statistically significant, it may be premature to propose SHARPIN as a biomarker.

R: Thank you very much for the comment; we added in the discussion that further analysis is required to suggest SHARPIN as a Biomarker.

  1. Provide quantitative (densitometric) analysis for the Western blots in Figure 4 to strengthen the protein expression results.

R: Densitometric analyses were added

  1. When discussing delphinidin 3,5-diglucoside losing efficacy after 48 hours, please include references to relevant literature that support the notion of compensatory pathways.
  2. We add relevant literature supporting these possible compensatory mechanisms regulating MGMT expression.

Reviewer 3 Report

Comments and Suggestions for Authors

The objectives here were defined by the authors: “We investigated the potential TMZ-sensitizing effect of delphinidin 3 glucoside and delphinidin 3,5 di-glucoside in glioblastoma cells. Our study suggests that glycosylated delphinidins downregulate NF-κB/MGMT signaling, leading to the sensitization of glioblastoma cells to TMZ.”

The results in the manuscript showed that delphinidins inhibit NF-κB signaling, according to them a critical pathway for GB progression, chemoresistance, and MGMT expression. So, they proposed the potential of these compounds as adjuvants in the treatment of GB. The TMZ resistance in glioblastoma is a problem in the success of therapy, and it would be very relevant to establish a natural strategy to this resistance. However, the manuscript needs to be improved in the description and discussion of results. Viability data, for example, do not indicate that delphinidin 3,5 diglucoside has potent anticancer effects when used alone since it is effective only at high concentrations (180 and 240 µM). And, the final results are not sufficient to indicate that delphinidin 3 glucoside acted in synergy with temozolomide to decrease cell viability. The effects on cell viability and apoptosis are just a sum of those observed in the two treatments, not synergy.

So, delphinidine 3,5-diglucoside reduced cell viability just in very high concentration, but it is interesting the fact that SVGp12 and GBM38 cells were less sensitive to these compounds. 

I suggest reviewing the text, especially in the description and discussion of the results. And also, better define the objectives.

Author Response

The objectives here were defined by the authors: “We investigated the potential TMZ-sensitizing effect of delphinidin 3 glucoside and delphinidin 3,5 di-glucoside in glioblastoma cells. Our study suggests that glycosylated delphinidins downregulate NF-κB/MGMT signaling, leading to the sensitization of glioblastoma cells to TMZ.”

The results in the manuscript showed that delphinidins inhibit NF-κB signaling, according to them a critical pathway for GB progression, chemoresistance, and MGMT expression. So, they proposed the potential of these compounds as adjuvants in the treatment of GB. The TMZ resistance in glioblastoma is a problem in the success of therapy, and it would be very relevant to establish a natural strategy to this resistance. However, the manuscript needs to be improved in the description and discussion of results. Viability data, for example, do not indicate that delphinidin 3,5 diglucoside has potent anticancer effects when used alone since it is effective only at high concentrations (180 and 240 µM). And, the final results are not sufficient to indicate that delphinidin 3 glucoside acted in synergy with temozolomide to decrease cell viability. The effects on cell viability and apoptosis are just a sum of those observed in the two treatments, not synergy.

R: We sincerely appreciate this review; we made the following changes to the manuscript

We deleted the following sentence in the manuscript: “Notably, delphinidin 3,5 diglucoside shows a greater inhibitory capacity, reducing viability by up to 70% at concentrations of 180 μM, suggesting an anticancer activity dependent on the structure and degree of glycosylation of the molecule [27-28]”

We modified the word synergy for additive effect in the following text : “Combination experiments with temozolomide revealed a clear synergy between delphinidin 3-glucoside and temozolomide, significantly decreasing cell viability, which did not occur with the combination of temozolomide and delphinidin-3,5-diglucoside”

So, delphinidine 3,5-diglucoside reduced cell viability just in very high concentration, but it is interesting the fact that SVGp12 and GBM38 cells were less sensitive to these compounds.

R: We appreciate this review. This finding was very important to us, and we discussed it in the text.

I suggest reviewing the text, especially in the description and discussion of the results. And also, better define the objectives.

R: We made the respective modifications to the manuscript
